


# Observations for high-impact weather and their use in verification

Chiara Marsigli[1,2], Elizabeth Ebert[3], Raghavendra Ashrit[4], Barbara Casati[5], Jing Chen[6], Caio A. S. Coelho[7], Manfred Dorninger[8], Eric Gilleland[9], Thomas Haiden[10], Stephanie Landman[11], Marion Mittermaier[12]

[1]Deutscher Wetterdienst, Offenbach am Main, 63067, Germany
[2]Arpae Emilia-Romagna, Bologna, 40122, Italy
[3]Bureau of Meteorology, Docklands, Victoria, 3008, Australia
[4]National Centre for Medium Range Weather Forecasting (NCMRWF), Noida, 201307, India
[5]MRD/ECCC, Dorval (QC), H9P 1J3, Canada
[6]Center of Numerical Weather Prediction, CMA, Beijing, 100081, China
[7]Centre for Weather Forecast and Climate Studies, National Institute for Space Research, Cachoeira Paulista, 12630-000, Brazil
[8]University of Vienna, Vienna, 1090, Austria
[9]Research Applications Laboratory, National Center for Atmospheric Research, Boulder, 80301, Colorado, U.S.A.
[10]ECMWF, Reading, RG2 9AX, UK
[11]South African Weather Service, Pretoria, 0001, South Africa
[12]MetOffice, Exeter, EX1 3PB, UK

*Correspondence to*: Chiara Marsigli (chiara.marsigli@dwd.de)

**Abstract.** Verification of high-impact weather is needed by the Meteorological Centres, but how to perform it still presents many open questions, starting from which data are suitable as reference. This paper reviews new observations which can be considered for the verification of high-impact weather, and provides advice for their usage in objective verification. Two high-impact weather phenomena are considered: Thunderstorm and fog. First, a framework for the verification of high-impact weather is proposed, including the definition of forecast and observations in this context and creation of a verification set.
Then, new observations showing a potential for the detection and quantification of high-impact weather are reviewed, including remote sensing datasets, products developed for nowcasting, datasets derived from telecommunication systems, data collected from citizens, reports of impacts and claim/damage reports from insurance companies. The observation characteristics which are relevant for their usage in forecast verification are also discussed. Examples of forecast evaluation and verification are then presented, highlighting the methods which can be adopted to address the issues posed by the usage of these non-conventional
observations and objectively quantify the skill of a high-impact weather forecast.

## 1 Introduction

Verification of forecasts and warnings issued for high-impact weather is increasingly needed by operational centres. The model and nowcast products used in operations to support the forecasting and warning of high-impact weather such as thunderstorm cells also need to be verified. The World Weather Research Programme (WWRP) of the World Meteorological Organisation





(WMO) has launched in 2015 the High-Impact Weather Project (HIWeather), a 10-year international research project, which will advance the prediction of hazards weather-related (Zhang et al., 2019). Forecast evaluation is one of the main topics of the project. The WWRP/WGNE Joint Working Group on Forecast Verification Research (JWGFVR)[1] of WMO has among its main tasks to facilitate the development and application of improved diagnostic verification methods to assess and enable improvement of the quality of weather forecasts. In the context of high-impact weather forecast, one the main aims is to

encourage the development of verification approaches making use of new sources and types of observations. The verification of high-impact weather requires a different approach than the traditional verification of the meteorological variables (for example, precipitation, temperature, wind) constituting the ingredients of the high-impact weather phenomenon. The phenomenon should be verified with its spatio-temporal extent and by evaluating the combined effect of the different meteorological variables that constitute the phenomenon.

Verification of weather forecasts is often still restricted to the use of conventional observations such as surface synoptic observations (SYNOP) reports. These conventional observations are considered the gold standard with well defined requirements for where they can be located and their quality and timeliness (WMO-No.8, 2018). However, for the purposes of verifying forecasts of high-impact weather, these observations often do not permit characterization of the phenomenon of interest, and therefore do not provide a good reference for objective verification. In Europe, ten years ago, a

list of new products to be subject to routine verification was proposed by Wilson and Mittermaier (2009) following the Member States and Co-operating States´ user requirements for ECMWF products. Among others, visibility/fog, atmospheric stability indices and freezing rain were mentioned, and the observations needed for the verification of these additional forecast products were reviewed.

Depending on the phenomenon, many reference data sets exist. Some are direct measurements of quantities to verify, e.g.

lightning strikes compared to a lightning diagnostic from a model, but many are not. In that case, we can derive or infer estimates from other measurements of interest. The options are many and varied, from remote sensing datasets, datasets derived from telecommunication systems including cell phones, data collected from citizens, reports of impacts and claim/damage reports from insurance companies. In this instance, it enables the definition of "observable" quantities which are more representative of the severe weather phenomenon (or its impact) than, for example, purely considering the accumulated

precipitation for a thunderstorm. These less conventional observations, therefore, enable more direct verification of the phenomena (e.g. a thunderstorm) and not just the meteorological parameters combining to determine their occurrence.

The purpose of this paper is to present a review of new observations, or more generically, quantities which can be considered as reference data or proxies, which can be used for the verification of high-impact weather phenomena. Far from being exhaustive, this review seeks to provide the numerical weather prediction (NWP) verification community with an organic

"starter-package" of information about new observations which may be suitable for high-impact weather verification, providing at the same time some hints for their usage in objective verification. In this respect, in this paper the word

---

[1] The JWGFVR is a Working Group joint between WWRP and WGNE (Working Group on Numerical Experimentation).



"observations" will be used interchangeably with the word "reference data", considering also that in some cases what is usually considered an observation may be only a component to build the "reference data" against to which verify the forecast (for example, a measurement of lightning with respect to a thunderstorm cell).

The review is limited to the "new" observations, meaning those observations which are not already commonly used in the verification of weather forecasts. The choice of what is a new observation is necessarily subjective. For example, we have chosen to consider radar reflectivity as a "standard" observation for the verification of precipitation, while the lightning measurements are considered among the "new" observations. Some of the "new" observations are familiar to meteorologists (e.g. for monitoring and nowcasting) but quite new for the NWP community, particularly from the point of view of their usage

in forecast verification. Though this paper focuses on thunderstorms and fog, it is possible to extend the subject to a wider spectrum of high-impact weather phenomena following the same approach.

## 2 A proposed framework for High Impact Weather verification using non-conventional observations

In order to transform the different sources of information about high-impact weather phenomena into an "objective reference" against which to compute the score of the forecast, four steps are required.

The first step is to define the quantity or object to be verified, selected for forecasting the phenomenon, which will be simply referred to as "forecast", even if it may not be a direct model output or a meteorological variable. As noted above, for thunderstorms, accumulated precipitation may not be the only quantity to be objectively verified, but it can be a component of the entity to be verified, along with lightning, strong winds and hail. Suitable quantifiable components should be directly observable or have a proxy highly correlated to it. In the case of thunderstorms, the quantity to be verified can be the lightning

activity, or areas representing the thunderstorm cells. An example of the latter case is given by the cells predicted by the algorithms developed in the context of nowcasting, where the cells are first identified in the observations from radar or satellite data, possibly in combination with other sources of data. With Doppler-derived wind fields, the occurrence of damaging winds could also be explored. In the case of fog, the quantity to be verified can be visibility or fog areas, either directly predicted by a model or obtained with a post-processing algorithm.

The second step is to choose the "observation," or reference, against which to verify the forecast. Observations which represent some of the phenomena described above already exist, but often they are used only qualitatively, for the monitoring of the events in real-time or for nowcasting. Ideally, these new observation types should have adequate spatial and temporal coverage, and their characteristics and quality should be well understood and documented. Even when a forecast of a "standard" parameter (e.g. 2m temperature) is verified against a "standard", conventional, observation (e.g. from SYNOP), care should

be taken in establishing the quality of the observations used as reference and their representativeness of the verified parameter. This need becomes stronger in case of non-standard parameters (e.g. a convective cell) verified against a non-standard observation (e.g. lightning occurrence or a convective cell "seen" by some algorithm). In order to indirectly assess the quality of the observations, or to include the uncertainty inherent in them, comparing observed data relative to the same parameter but



coming from different sources is a useful strategy. This approach can contribute to increasing the different sources of
"observations" available for the verification of a phenomenon, considering that they are all uncertain. In the verification of
phenomena, when new observations are used for verifying new products, it becomes even more crucial to include the
uncertainties inherent in the observations in the verification process, as they will affect the objective assessment of forecast
quality in a context in which there may not be a previously viable evaluation of that forecast. How to include uncertainties in
verification will not be discussed in the present paper. For a review of the current state of the research, it is suggested to read
Ben Bouallegue et al. (2020) and the references therein.

After having identified the forecast and the reference, the third step is the creation of the pair, called a verification set. The
matching of the two entities in the pair should be checked before the computation of summary measures. For example, is one
lightning strike sufficient for verifying the forecast thunderstorm cell? Since "forecast" and "observation" do not necessarily
match in the context of high impact weather verification using non-conventional observation, a preparatory step is needed for
ensuring a good degree of matching. In this step, the correlation between the two components of the pair should be analysed.
Some of the observation types may be subject to biases. As correlation is insensitive to the bias, for some types of forecast-
observation pairs, any thresholds used to identify the objects of the two quantities must also be studied to ensure that the
identification and comparison is as unbiased (from the observation point-of-view) as possible. In particular, the forecast and
the observation should represent the same phenomenon, and this can be achieved by stratifying the samples. A simple example
is the case where the forecast is "a rainfall area" and the observation is "a lightning strike": all the cases of precipitation not
due to convection in the forecast sample will make the verification highly biased, therefore they should be excluded. Another
element of the matching is to assess spatial and temporal representativeness, which may lead to the need to suitably average
or re-grid the forecasts and/or observations. Some examples of how the matching is performed are presented in the next
Sections. If the forecasts and observations show a good statistical correlation (or simply a high degree of correspondence), it
can be assumed that one can provide the reference for the other and objective verification can be performed. This approach
can be extended to probabilities: an area where the probability of occurrence of a phenomenon exceeds a certain threshold can
be considered as the predictor for the forecast of the phenomenon, provided that its quality as predictor has been established
through previous analysis. Therefore, the same verification approach can be applied in the context of ensemble forecasting
(Marsigli et al., 2019). Verification of probability objects for thunderstorm forecast is performed in Flora et al. (2019).

A special case of pair creation occurs when an object identified by an algorithm is used as the reference, as in the example of
the thunderstorm cell identified by radar. In this case, a choice in the matching approach is required: should the algorithm be
applied only for the identification of the phenomenon as observation, or should it also be applied to the model output for
identifying the forecast phenomenon? This is similar to what is done in standard verification, when observations are upscaled
to the model grid, to be compared to a model forecast, or instead both observations and model output are upscaled to a coarser
grid. In the first case, the model forecasts, expressed as a model output variable (e.g. the precipitation falling over an area), is
directly compared with the "observation" (e.g. a thunderstorm cell identified by an algorithm on the basis of some
observations). In the second case, the same algorithm (an observation operator) is applied to the same set of meteorological



variables in both the observed data and the model output in order to compare homogeneous quantities. This approach eliminates some approximations made in the process of observation product derivation, but observation operators are also far from perfect.

Therefore, although this approach can ensure greater homogeneity between the variables, it may still introduce other errors resulting from the transformation of the model output.

Finally, the fourth step is the computation of the verification metrics. This step is not principally different from what is usually done in objective verification, taking into account the specific characteristics of the forecast/observation pair. In general, high-impact weather verification requires an approach to the verification problem where the exact matching between forecast and

observation is rarely possible, therefore verification naturally tends to follow fuzzy and/or spatial approaches (Ebert, 2008; Dorninger, 2020). An issue inherent in the verification of objects, such as convective cells, is the definition of the "non-event": while it is intuitive how to perform the matching between a forecasted and an observed convective cell, how to define the mismatch between a forecasted cell and the non-occurrence of convective cells in the domain (false alarms) is not trivial. This question should be addressed when the verification methodology is designed and the answer may depend on the specific

methodology adopted for the spatial matching.

## 2 High-impact weather: Thunderstorms

In convective situations, meteorological centres use a large amount of real-time data from different sources (e.g. ground-based, satellite, and radar) in order to perform nowcasting and monitoring of thunderstorms and thus issue official weather warnings. In recent years, with the development of convection-permitting models, thunderstorm forecasting has become an aim of short-

range forecasting, particularly with the advent of modeling frameworks such as the Rapid Update Cycle (RUC), which provides NWP-based forecasts at the 0-6h time scale. These predictions of thunderstorms, from nowcasting to forecasting, need to be verified in order to provide reliable products to the forecasters and to the users.

New observations which could be used, or have been used on a limited basis, for the verification of thunderstorms, are reviewed here, categorized as lightning detection networks, nowcasting products and data collected through human activities. Examples

of usage of these data, sometimes in combination, in the objective verification of thunderstorms are presented.

### 2.1 Lighting detection networks

Data from lightning detection networks are used for nowcasting purposes in several centres. Lightning density and its temporal evolution can serve as a useful predictor for the classification of storm intensity and its further development (Wapler et al, 2018). Therefore, these data show a good potential in thunderstorm verification. Lightning data can be used as observations in

different ways, from the most direct, verifying a forecast also expressed in terms of lightning, to more indirect, for example, by verifying a predicted thunderstorm cell. In both cases, some issues need to be addressed, starting from how many strokes are needed to detect the occurrence of a thunderstorm (specification of thresholds). In the indirect cases, how large an area defines the region of the phenomenon (thunderstorm) needs to be specified. These choices are relevant also when lightning



data are combined with other observations (e.g. radar data), in order to improve the detection of the phenomenon. Some of
these issues are further discussed when examples of forecast verification against lighting data are presented.

Data from lightning localisation networks have the advantage of continuous space and time coverage and of a high detection
efficiency, compared to human thunderstorm observations at specific stations. Some lightning detection networks which have
been used for thunderstorm verification (often only subjectively) are listed in Table 1.

| Name of the dataset | Scope | Short Description | References |
| --- | --- | --- | --- |
| EUCLID (EUropean Cooperation for LIghtning Detection) | Collaboration among national lightning detecting networks over Europe | Lightning data with homogenous quality in terms of detection efficiency and location accuracy. About 164 sensors in 27 countries. | www.euclid.ord |
| LINET (LIghtning detection NETwork) | Originally developed at the University of Munich | Lightning sensors set up in the area to be monitored (baseline 200 to 250 km). Information about location, time and stroke current. | www.nowcast.de Betz et al. (2009) |
| ATDnet (Arrival Time Difference network) | Met Office | Network of 11 sensors around the world. | https://navigator.eumetsat.int/ product/EO:EUM:DAT:OBS: ATDNET Anderson and Klugmann (2014) |
| LAMPINET (lampi means lightnings in Italian) | Meteorological Service of the Italian Air Force | Provides the number and the intensity of the strokes. | Biron (2009) |
| NORDLIS (Nordic Lightning Information System) | Scandinavia | Cooperative lightning location network between Norway, Sweden, Finland and Estonia; about 30 sensors. | Mäkelä et al. (2010) |
| National Lightning Detection Network (NLDN) | USA | NOAA develops derived products freely available for all users, including summaries of lightning flashes by county and state and gridded lightning frequency products. | Cummins and Murphy (2009) https://www.ncdc.noaa.gov/data-access/severe-weather/lightning-products-and-services |





| SA-LDN (South African Lightning Detection Network) | South African Wether Service | Network of 26 Vaisala cloud-to-ground lightning detection sensors | Gijben (2012) |
|---|---|---|---|

Table 1. Lightning detection networks used for thunderstorm verification in the papers referenced in this Section.

Lightning observations are also provided from space (Table 2), which can complement ground-based weather radars over sea and in mountainous regions.

| Sensor/dataset | Operated by | Short Description | References |
|---|---|---|---|
| Lightning Imaging Sensor (LIS) | International Space Station (ISS) | Provides total lightning measurements between +/- 48 degrees latitude. | Blakeslee and Koshak, 2016 |
| Geostationary Lightning Mapper (GLM) | GOES-16 (NOAA) | Measures a region including the United States, providing lightning detection with a spatial resolution of about 10 km. | https://ghrc.nsstc.nasa.gov/lightning |
| Geostationary Lightning Imager (or Lightning Mapping Imager) | FY-4 (CMA) | Provides measurements of the total lightning activity with a resolution of about 6 km at the subsatellite point. | https://fy4.nsmc.org.cn/nsmc/en/theme/FY4A_instrument.html#LMI |
| Lightning imager mission | Meteosat Third Generation | Provides lightning products with 4.5 km resolution. | www.eumetsat.int |

Table 2. Lightning sensors on board satellites and the International Space Station.

## 2.2 Nowcasting products

National Meteorological Services develop tools for nowcasting, where observational data from different sources (satellite, radar, lightning, …) are integrated in a coherent framework (Wapler et al., 2018 and Schmid et al. 2019), mainly with the purpose of detecting and very short range prediction of high-impact weather phenomena. In this context, the use of machine learning to detect the phenomenon from new observations, after training the algorithm with a sufficiently large sample of past observations, could play an important role by mimicking a decision tree where different ingredients (predictors) are combined and computing how they should be weighted relatively. Usually different algorithms are developed for the different products. For a description of nowcasting methods and systems, see WMO-No.1198 (2017) and Schmid et al. (2019).



For the purpose of this paper, the detected variables/objects of nowcasting (thunderstorm cells, hail, etc.) can become observations against which to verify the model forecast. The detection step of the nowcasting algorithm can be considered as
a sort of "analysis" of the phenomenon addressed by that algorithm, e.g. an observation of a convective cell, which could be used for verifying the forecasts of this phenomenon. Here, nowcasting products are proposed as observed data instead of prediction tools.

Remote-sensing-based nowcasting products have the clear advantage of offering high spatial continuity over vast areas. As a disadvantage, it should be noted that some data have only a qualitative value, but qualitative evaluation could become
quantitative by "relaxing" the comparison through neighbourhood/thresholding (examples will be provided in Section 2.4). The quantification of the errors affecting the products is also an issue, since usually this is not provided by the developers. Efforts towards such an error quantification should be made to provide appropriate confidence in the verification practice.

Exploring the possible usage of the variables/objects identified through nowcasting algorithms for the purpose of forecast verification requires strengthening the collaboration between the verification and the nowcasting communities. On the one
hand, the nowcasting community has a deep knowledge of the data sources used in the nowcasting process and of the quality of the products obtained by their combination. On the other hand, the forecast verification community can select, from the huge amount of available data, those showing greater reliability and offering a more complete representation of the phenomenon to be verified.

Many meteorological centres have developed their own nowcasting algorithms, (Wapler et al., 2019), usually different for
different countries, but a complete description of them is difficult to find in the literature. Therefore, it is recommended to contact the Meteorological Centre of the region of interest. The example of EUMETSAT collaboration shown below indicates the kind of available products.

### 2.2.1 The EUMETSAT NWC-SAF example

EUMETSAT (www.eumetsat.int) proposes several algorithms for using their data. The Satellite Application Facility for
supporting NoWCasting and very short range forecasting (NWC-SAF, nwc-saf.eumetsat.int) provides software packages for generating several products from satellite: clouds, precipitation, convection and wind (Ripodas et al., 2019). The software is distributed freely to registered users of the meteorological community. The European Severe Storms Laboratory (ESSL, https://www.essl.org) has performed a subjective evaluation of the products of the NWC-SAF for convection (Holzer and Groenemeijer, 2017) indicating the usefulness of the products for nowcasting and warning for objective verification. For
example, the stability products (Lifted Index, K-index, Showalter Index) and Precipitable Water (low, mid, and upper troposphere) have been judged to be of some value. The RDT (Rapid Developing Thunderstorms) product in its present form has been judged difficult to use, but elements of the RDT product have significant potential for the nowcasting of severe convective storms, namely the cloud-top temperatures and the overshooting-top detections. Furthermore, the RDT index has proven to be useful in regions where radar data do not have full coverage. De Coning et al. (2015) found good correlations
between the storms identified by the RDT and the occurrence of lightning over South Africa. Gijben and de Coning (2017)




showed that the inclusion of lightning data had a positive effect on the accuracy of the RDT product, when compared to radar data, over a sample of twenty-five summer cases. RDT is considered as an observation against which to compare model forecasts of convection, as well high impact weather warnings, at the South African Weather Service (S. Landman, personal communication). A review of the products for detecting the convection is made by the Convection Working Group, an initiative

of EUMETSAT and its member states and ESSL (https://cwg.eumetsat.int/satellite-guidance, Fig. 1), indicating which products are appropriate for the detection of convection.

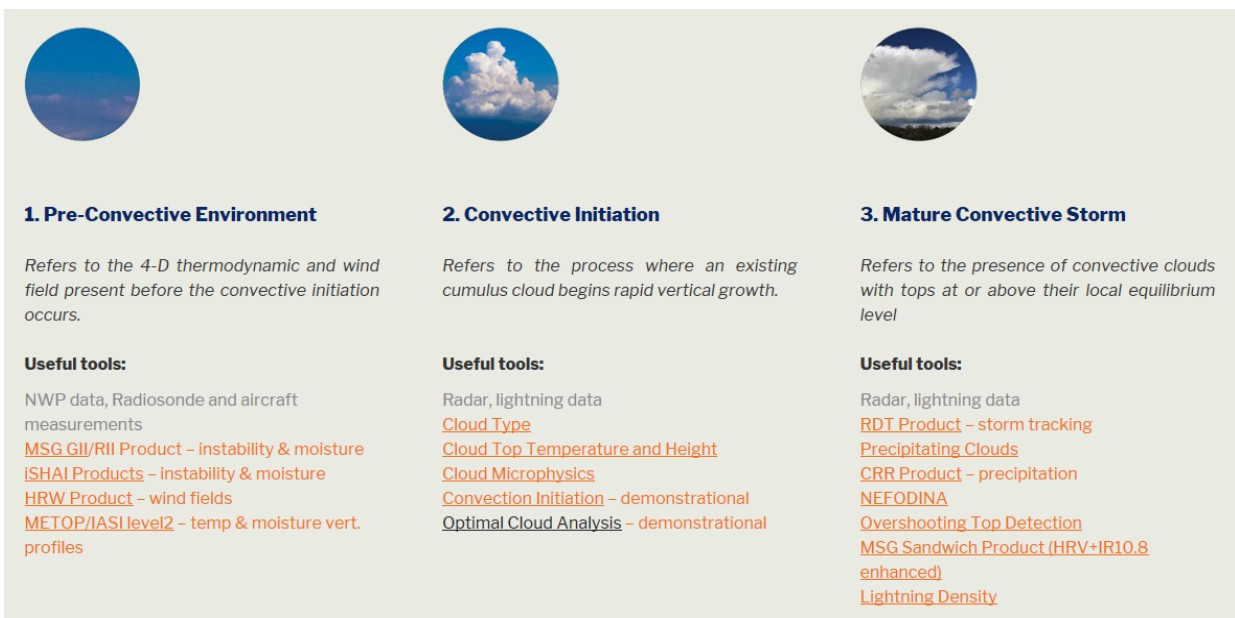

**Figure 1. Satellite-derived products for the detection of convection. From https://cwg.eumetsat.int/satellite-guidance.**

## 2.3 Data collected through human activities

Human activities permit generation and collection of a wide and diverse amount of data about the weather. Some of these data are collected with the purpose of monitoring the weather (citizen networks, reports of severe weather), others are generated for quite different purposes but may be related to the weather (impact data, insurance data). A special category are the data generated in the social networks: a report of severe weather may be generated only for the purpose of complaining about the weather. The quality of these data and their correlation with weather phenomena are very different in these three cases. A

review of the so-called "volunteered geographic information" related to weather hazards has been performed by Harrison et al. (2020), including also crowdsourcing and social media.

Data collected through human activities can be distinguished by those solicited by the potential user and those that are not. An example of the first is report data collected in a dedicated website, prepared by the potential user, inviting the citizen to submit their report. Table 3 describes three widely used databases. This kind of data may be biased by the fact that people may feel

pushed to report. Non-solicited data include those associated with public assistance (e.g. emergency services) and those



spontaneously generated by people (e.g. in social networks). Solicited and un-solicited human reports need to be quality checked in order to take into account and correct for biases which are introduced (e.g. multiple reports of the same event with little spatial or temporal distance, subjective and conflicting evaluations of the intensity, variation of the sample with time, etc.). Human reports and impact data have the advantage of being particularly suitable for high impact weather verification,

and the disadvantage that they are biased according to the density of the population: where no or few people live, no reports are issued even if the impact on the environment may be high. Therefore, this kind of data has a particularly inhomogeneous spatial distribution.

| Dataset Name | Operated by | Characteristics | References |
|---|---|---|---|
| European Severe Weather Database (ESWD) | ESSL (European Severe Storm Laboratory) | Quality-controlled information on severe convective storm events over Europe. Among other parameters, hailstones with a diameter of at least 2.0 cm are reported. | https://www.eswd.eu Dotzek et al., 2009 |
| Storm Prediction Center (SPC) | NOAA | Archive of severe weather events, including tornadoes, wind, hail, thunderstorms. | https://www.spc.noaa.gov |
| Severe Storms Archive | BoM (Australian Bureau of Meteorology) | Data relating to recorded severe thunderstorm and related events in Australia dating back to the 18th Century. Information on related severe weather (e.g. wind gusts) is also provided. | http://www.bom.gov.au/australia/stormarchive |

**Table 3. Widely used severe weather report databases.**


Severe weather reports are an important source of information which can be made objective and used in forecast verification; they can be also be combined with other data sources to identify the phenomenon. Valachova and Sykorova (2017 and 2019) use a combination of satellite data, lightning data and radar data (cells identified through a tracking algorithm) in order to detect thunderstorms and their intensity for nowcasting purposes, in combination with reports from ESWD operated by ESSL.

Crowdsourcing can also provide a new kind of data useful for verifying thunderstorm as a phenomenon, using the reports from the citizens as observations of the thunderstorm. Crowd-sourced hail reports gathered with an app from MeteoSwiss makes an extremely valuable observational dataset on the occurrence and approximate size of hail in Switzerland (Barras et al., 2019). The crowdsourced reports are numerous and account for much larger areas than automatic hail sensors, with the advantage of



unprecedented spatial and temporal coverage, but provide subjective and less precise information on the true size of hail.

Therefore, they need to be quality controlled. Barras et al. (2019) noted that their reflectivity filter requires reports to be located close to a radar reflectivity area of at least 35 dbZ. Overall, the plausibility filters remove approximately half of the reports in the dataset. The strength as well as the weakness of these data resides in their being representative only of the weather in populated areas.

Among the impact data, insurance data are a useful source, but their availability is limited because of economic interests
(Pardowitz, 2018).

A special branch of non-solicited impact data are those generated in the social networks. Recent work at Exeter University has shown that social sensing provides robust detection/location of multiple types of weather hazards (Williams, 2019). In this work, twitter data were used for sensing the occurrence of flooding. Methods to automate detection of social impacts are developed, focusing also on the data filtering to achieve a good quality. In the usage of all these data as observations for an
objective verification, a crucial step is the pre-processing of the data in order to isolate the features which are really representative of the forecasted phenomenon. As noted earlier, an assessment of the observation error should accompany the data, for example by varying the filtering criteria and producing a range of plausible observations.

### 2.4 Usage of the new observations in the evaluation or verification of thunderstorms

In this section, studies describing the verification of thunderstorm and convection are presented, highlighting which
verification methods have been used and how the authors addressed the issues posed by the usage of non-standard observations and by the need to create a meaningful verification set. The different observations listed in the previous subsection are used, sometimes in combination. For each work, their most significant feature for the purpose of this paper is indicated in the title as keyword(s).

*Lightning; matching forecast and observation.* Caumont (2017) verified lightning forecasts by computing 10 different proxies for lightning occurrence from the forecast of the AROME model run at 2.5 km. In order for them to be good proxies for lightning, calibration of their distribution was performed, a good example of the process of matching the predicted quantity with the observed one.

*Lightning; spatial coverage.* For a thunderstorm probability forecasting contest, Corbosiero and Galarneau (2009) and
Corbosiero and Lazear (2013) predicted the probability to the nearest 10% that a thunderstorm was reported during a 24-hour period at ten locations across the continental United States. The forecasts were verified by standard METAR reports as well as by the National Lightning Detection Network (NLDN) data. For this comparison, strikes were counted within 20 km of each station (Bosart and Landin, 1994). Results show that, although there was significant variability, the 10-km NLDN radial ring best matches METAR thunderstorm occurrence.

*Lightning; spatial coverage; pointwise vs spatial.* Wapler et al. (2012) computed the Probability of Detection (POD) by comparing the cells detected by two different nowcasting algorithms to lightning data. The comparison was made both





pointwise and by using an areal approach (following Davis et al., 2006). They showed how the score improves by increasing the radius of the cell and that the verification is also dependent on the reflectivity threshold used to detect the cells. The pointwise approach gave a higher P than the areal approach, showing how the choice of the verification method influences the results. The area covered by lightning generally extends beyond the area covered by a cell (identified by high reflectivity values), leading to a decrease of the areal POD. They also compared detection of hail from the ESWD report with the cells detected by two nowcasting algorithms, but no scores were computed. As the authors pointed out, since observations of "no event" are not provided, it is difficult to compute scores resulting from the presence of "yes-event" observations only.

*Lightning; spatial coverage; false alarms.* Lighting data were used for verification of nowcasting of thunderstorms from satellite data by Müller et al. (2017). They used a search radius of 50 km, within which at least 2 flashes in 10 minutes should be recorded in order to detect the event. A false alarm was identified when the nowcasted thunderstorm was not associated with a detected event.

*Lightning; spatial method.* In the work of de Rosa et al. (2017) extensive verification of thunderstorm nowcasting was performed against ATDNet lightning data over a large domain covering central and southern Europe using the MODE method (Davis, 2006; Bullock et al., 2016) of the MET (Model Evaluation Tools) verification package. The cells detected from the nowcasting algorithm were compared against lightning objects obtained by clustering the strikes.

*Combination of lightning and report data.* Wapler et al. (2015) performed a verification of warnings issued by the DWD for two convective events over Germany. They qualitatively compared warning areas against data from ESWD Reports and lightning data. They also performed a quantitative comparison against lightning data via a contingency table for the two events. This approach could be extended to a larger dataset in order to perform a statistically robust verification.

*Combination of lightning and report data.* At ECMWF, two parameters, convective available potential energy (CAPE) and a composite CAPE–shear parameter, have recently been added to the Extreme Forecast Index / Shift Of Tails products (EFI/SOT), targeting severe convection. Verification is performed against datasets containing severe weather reports only and a combination of these reports with ATDNet lightning data (Tsonevsky et al., 2018). Verification results based on the area under the relative operating characteristic curve show high skill at discriminating between severe and non-severe convection in the medium range over Europe and the United States (Fig. 2). More generally, report data and lightning data were used to evaluate the performance of ECMWF systems in a collection of severe-weather related case studies, including convection, for the period 2014-2019 (Magnusson, 2019).


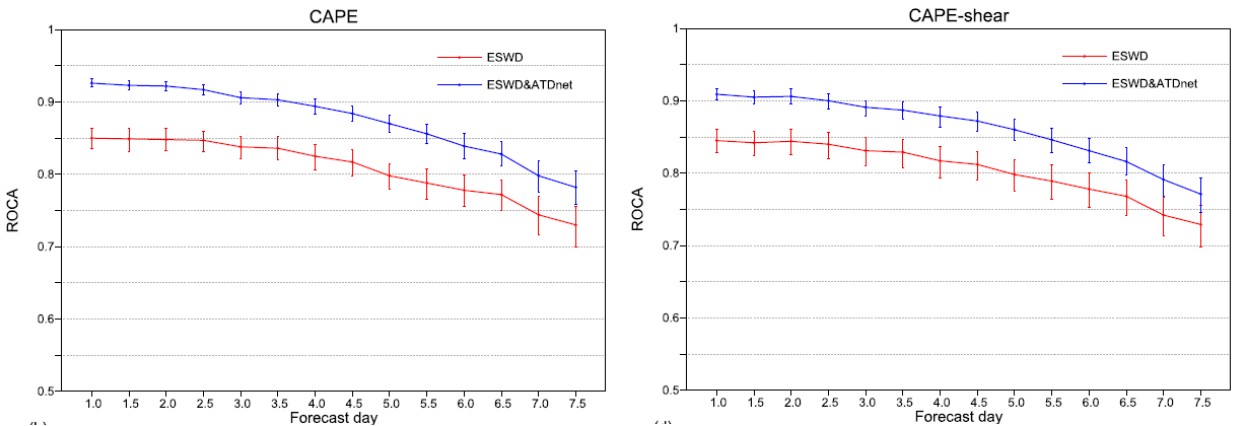


**Figure 2. Area under the ROC Curve for EFI for CAPE (left) and CAPE-shear (right) for Europe, compared against datasets containing severe weather reports only (red curves) and a combination of these reports with ATDNet lightning data (blue curves). From Tsonevsky et al. (2018).**

*Satellite data; observation operator.* Wilson and Mittermaier (2009) employed a MSG satellite derived Lifting Index to evaluate the Lifted Index forecasted by the model, to verify regions of convective activity. The index can be computed only on cloud-free areas but, with respect to that derived from radiosoundings, has the advantage of much larger geographical coverage.

*Satellite data; observation operator.* Keller et al. (2015) and Rempel et al. (2017) performed a direct comparison of cloud 325 properties between the convection-permitting model and the satellite data using an observation operator to derive synthetic satellite images from the model.

*Satellite data; matching forecast and observation.* Deep convection in a convection-permitting model, compared to the one forecasted by a convection-parametrised model, was evaluated by Keller et al. (2015). The observables were satellite-derived Cloud Top Pressure, Cloud Optical Thickness, Brightness Temperature, Outgoing Longwave Radiation and Reflected Solar 330 Radiation. They were used in combination to evaluate the characteristics of the convection. No objective verification was performed, but a categorization of the satellite products (Fig. 3) permits the use of these indicators for verification of cloud type and timing of convection.

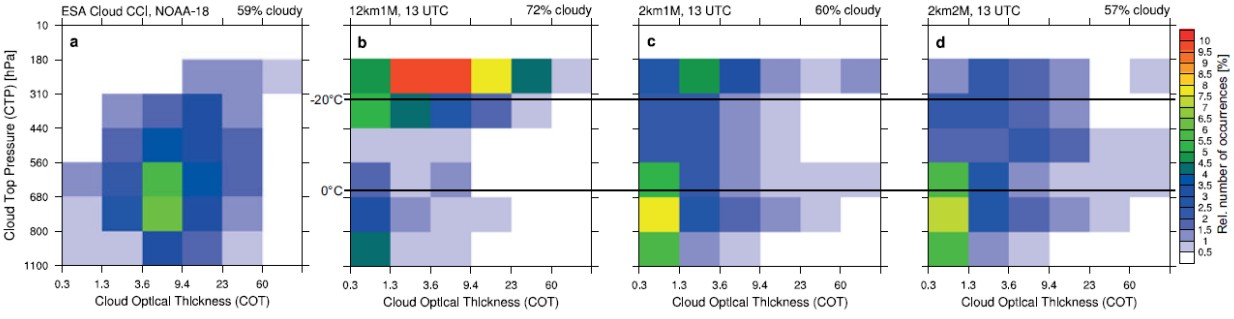



**Figure 3. Histograms of cloud frequency as a function of cloud optical thickness and Cloud Top Pressure for (a) satellite data and**
**(b,c,d) three simulations made with the COSMO model at 12 km (b) and 2 km (c,d, the latter with enhanced microphysics scheme)**
**horizontal resolution. From Keller et al. (2015).**

*Evaluation from citizen used for objective verification.* MeteoSwiss carried out a subjective verification by beta testers of
thunderstorm warnings issued for municipalities on mobile phones via app (Gaia et al., 2017). The forecast was issued in
categories: "a developing / moderate / severe / very severe thunderstorm is expected in the next XX minutes in a given
municipality". Probability of Detection and False Alarm Rate of the thunderstorm warnings, computed against beta-tester
evaluation, were computed.

*Insurance data and emergency call data.* Schuster et al. (2005) analysed characteristics of hailstorms using data of insurance
claims costs, emergency calls and requests of fire brigade intervention, in addition to data from severe weather reports (Fig.
4). Rossi et al. (2013) used weather-related emergency reports archived in a database by the Ministry of the Interior in Finland
to determine hazard levels for convective storms detected by radar, demonstrating the potential of these data especially for
long-lasting storms.

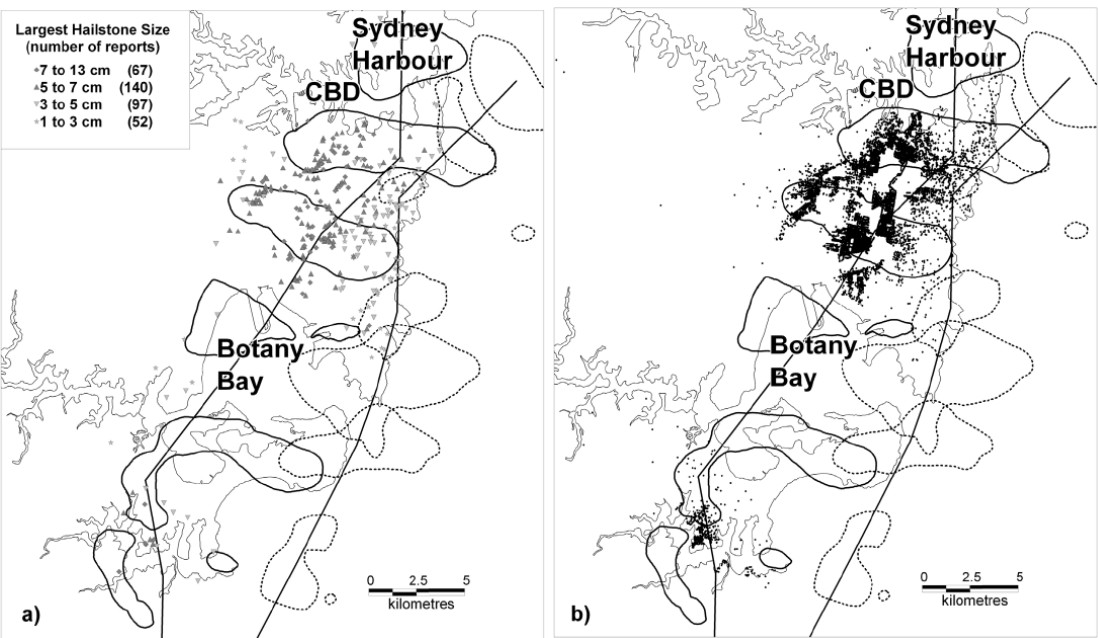

**Figure 4. Affected area and storm paths derived from radar data (closed contours) with (a) hailstone sizes categorized according to**
**their diameters in cm and (b) requests for assistance to the NSW State Emergency Service. From Schuster et al. (2005).**

*Emergency services data; matching forecast and observation.* Pardowitz and Göber (2017) compared convective cells detected
by a nowcasting algorithm against data of fire brigade operations. In addition to the location and time of the alerts, the data
included keywords associated to each operation, which permitted selection of the weather related operations relevant for this





work (water damages and tree related incidents). Pardowitz (2018) examined fire brigade operation data in the city of Berlin with respect to their correlation to severe weather.

*Airport operation data used in verification; matching forecast and observation.* A simple and effective verification framework for impact forecasts (in this case, probabilistic forecasts of thunderstorm risk) was demonstrated by Brown and Buchanan (2018) based on two years of data. The London Terminal Maneuvering Area (TMA) thunderstorm risk forecast is a specific

customer oriented forecast with the purpose of providing early warning of convective activity in a particular area. The London TMA thunderstorm risk forecasts were verified for a range of thresholds against observations provided by the UK Air Navigation Service Provider (ANSP) of the delay in minutes (arrival, en-route and alternative combined) and arrival flow rate experienced at airports across the London TMA. Applying thresholds to the delay in minutes and flow rate data received directly from the UK ANSP allows for categorization of each forecast period into a high or medium impact event. This process

enables a simple $2 \times 2$ contingency approach to be taken when verifying the forecasts. Brown and Buchanan (2018) analysed the results using relative operating characteristic (ROC) curves, reliability diagrams and economic value plots (e.g. Jolliffe and Stephenson, 2012).

## 3 High-impact weather: Fog

The second high-impact weather phenomenon considered here is fog. According to the WMO definitions, fog is detected in

observations when the visibility is below 1000 m, but in the context of high-impact weather different thresholds are often adopted, specific to the application of interest. NWP models generally are not efficient in predicting fog and visibility conditions near the surface (Steeneveld, 2015), since the stable boundary layer processes are typically not represented well in the NWP models. Therefore, for forecasting of fog/visibility, very-high-resolution (300 m grid) models are employed in some of the NWP centres (London Model in Met Office UK and Delhi Model in NCMRWF, India).

### 3.1 Observations for fog

Fog and low stratus can be detected by surface or satellite observations. Problems of visibility measures from manual and automatic stations are described in Wilson and Mittermaier (2009). In principle, surface instruments can easily detect fog (e.g. visibility and cloud base height measurements). However, observation sites are sparsely distributed and do not yield a full picture of the spatial extent of fog. Satellite data can compensate because of the continuous spatial coverage they provide

(Cermak and Bendix, 2008). The disadvantage of satellite data is the lack of observations when mid- and high-level clouds are present (Gultepe, 2007). Also, the precise horizontal and vertical visibility at the ground is difficult to assess based solely on satellite data. Satellite data alone cannot distinguish between fog and low stratus, because it cannot be determined if the cloud base reaches the surface. Therefore, detection methods usually include both phenomena. Satellite data are used for the detection of fog in several Weather Centres during the monitoring phase of operational practice. As an example, in India, fog is monitored

using satellite maps from fog detection algorithms developed for INSAT 3D satellite joint ISRO-Indian Space Research



Organization. Published evaluations of model outputs for the prediction of fog employing satellite images have mainly taken a case study approach (e.g. Müller et al., 2010; Capitanio, 2013). Only few studies employed satellite data in an objective verification framework, using spatial verification methods.

### 3.1.1 The EUMETSAT NWC-SAF example

The Satellite Application Facility for supporting NoWCasting and very-short-range forecasting (NWC-SAF) of EUMETSAT provides cloud mask (CMa) and cloud type (CT) products. In CMa, a multispectral thresholding technique is used to classify each grid point. The thresholds are determined from satellite-dependent look-up tables using as input the viewing geometry (sun and satellite viewing angles), NWP model forecast fields (surface temperature and total atmospheric water vapour content) and ancillary data (elevation and climatological maps). With the CT mask a cloud type category is attributed to the clouds

detected by the CMa. To assess the cloud height, the brightness temperature of a cloud at 10.8 μm is compared to a NWP forecast of air temperature at various pressure levels. The CMa followed by the CT application classifies each grid point as one of the listed categories (Derrien and Le Gléau, 2014). The points (or areas) that are labelled as fog can be used quantitatively, for example as a dichotomous (fog/no fog) value.

The 24h Microphysics RGB (red-green-blue; description available at www.eumetrain.org) makes use of three window

channels of MSG: 12.0, 10.8 and 8.7 μm. This product has been tuned for detection of low-level water clouds and can be used day and night throughout the year. In the 24h Microphysics RGB product fog and low clouds are represented by light greenish/yellowish colours. Transparent appearance (sometimes with gray tones) indicates a relatively thin, low feature that is likely fog. In order to use this product for fog and low cloud detection as an observation in a verification process, a mask could be created, where pixels with light greenish/yellowish colours are set to the value of 1 and all other colours set to 0.

Other Centres developed different algorithms for similar purposes: NOAA developed GEOCAT (Geostationary Cloud Algorithm Test-bed), which, among others, provide estimates fog probability from satellite. Thresholds should be applied to the probabilities in order to perform verification of model fog forecasts.

### 3.2 Usage of the new observations in the evaluation or verification of fog

Studies demonstrating the verification of fog forecasts are briefly described below. Depending on the observation they use and

on the purpose of the verification, point-wise or spatial verification methods are adopted. In addition, there is a distinction, depending on the purpose of verification, between model-oriented and user-oriented methods. Spatial verification, e.g. using objects, or allowing for some sort of spatial shift, can be informative for modelers and as a general guidance for users. If a user needs information about performance at a particular location only (e.g. an airport), it does not matter whether the fog that was missed in the forecast was correctly predicted at a nearby location. For this user, classical point-wise verification measures are

still the most relevant. However, with the increased use of probabilistic forecasts (to be recommended especially in the case of fog) this distinction blurs somewhat, because the 'nearby correct forecast' is likely to indirectly show up in the probabilistic scores of a given location. The work presented are organised as point-wise or spatial verification methods.



*Point-wise verification.*

Zhou and Du (2010) verified probabilistic and deterministic fog forecasts from an ensemble at 13 selected locations in China, where fog reports were regularly available. The forecast at the nearest model grid point was verified against observations over a long period. Boutle et al. (2016) performed an evaluation of the fog forecasted by a very-high-resolution run (333 m) of the Unified Model over the London area. Verification was made at three locations where measurements were available: two visibility sensors at two nearby airports and manual observations made at London City. Bazlova (2019) presented the results

of the verification of fog predicted with a nowcasting system at three airports in Russia, in the framework of the WMO Aviation Research Demonstration Project (AvRDP). Indices from the contingency table were computed against observations available from aviation weather observation stations.

Terminal Area Forecasts (TAFs) give information about the expected conditions of wind, visibility, significant weather and clouds at airports. Mahringer (2008) summarized a novel strategy to verify TAFs using different types of change groups, where

the forecaster gives a range of possible values valid for a time interval. A TAF thus contains a range of forecast conditions for a given interval. In this approach, time and meteorological state constraints are relaxed in verification. To evaluate the correctness of a forecast range, the highest (or most favourable) and lowest (or most adverse) conditions valid for each hour of the TAF are taken for verification. For this purpose, all observations within the respective hour are used (METAR and SPECI), which span a range of observed conditions. For each hour, two comparisons are made: the highest observed value is

used to score the highest forecast value and the lowest observed value is used to score the lowest forecast value. Entries are made accordingly into two contingency tables which are specific for weather element and lead time.

From a model-oriented verification point of view of fog and visibility, aiming at improving the modelling components, generally the focus is on how accurately the model reproduces ground and surface properties, surface layer meteorology, near surface fluxes, atmospheric profiles and aerosol and fog optical properties. Ghude et al. (2017) give details of an observation

field campaign for study of fog over Delhi in India. Such campaigns allow not only verification of various surface, near surface and upper air conditions, but also allow calibration of models, or the application of statistical methods to improve raw model forecasts at stations of high interest (e.g. airports). Micro-meteorological parameters like soil temperature and moisture, near surface fluxes of heat, water vapor and momentum in the very-high-resolution models (London Model and Delhi Model; 330m grid resolution) could be verified using the field data.


*Spatial verification using satellite data.*

Morales et al. (2013) verified fog and low cloud simulations performed with the AROME model at 2.5 km horizontal resolution using the object-oriented SAL measure (Wernli et al, 2008). The comparison was made against the Cloud Type product of NWC-SAF.

Westerhuis et al. (2018) and Ehrler (2018) used a different satellite-based method for fog detection. The detection method calculates a low-liquid-cloud confidence level (CCL method) from the difference between two infrared channels (12.0 μm and



8.7 µm) from Meteosat Second Generation with a spatial resolution of about 3 km. The grid points are also filtered for high- and mid-level clouds using the Cloud Type information from the NWC-SAF data. The CCL satellite data ensures consistent day- and nighttime detection of fog, in contrast to the NWC-SAF cloud detection, which was also tested as a reference. Cases
with high and mid-level clouds must be excluded. The same method for fog detection was also used in Hacker and Bott (2018) who studied the modeling of fog with the COSMO model in the Namib Desert. Ehrler (2018) verified the fog forecasted by the COSMO model using the Fractions Skill Score (Roberts and Lean, 2007). First, grid points with high- or mid-level clouds were filtered, both in the CCL satellite and in the COSMO model data. Then, the score was computed on the remaining points. The CCL satellite data is not binary, but rather ranges from 0 to 1, therefore an adequate CCL threshold above which a grid
point is assumed as fog needed to be determined. For this purpose, the FSS was calculated for different thresholds (0.5-0.8) in the CCL satellite data, leading to the choice of a threshold of 0.7.

In order to verify fog extent, measures considering the area impacted by a phenomenon could be employed, as for example the Spatial Probability Score (SPS) introduced by Goessling and Jung (2018).

## 4 Final considerations

This paper review non-standard observations and proposes that they be used in objective verification of forecasts of high-impact weather, focusing on thunderstorms and fog. Apart from the description of the data sources, their advantages and critical issues with respect to their usage in forecast verification are highlighted. Some verification studies employing these data, sometimes only qualitatively, are presented, showing the (potential) usefulness of the "new" observations for the verification of high-impact weather forecasts.

Several data sources are not "new" in other contexts, but have not been routinely applied for objective forecast verification. For example, some data are well established as sources of information for nowcasting and monitoring of high-impact weather. Others are used in the assessment of the impacts related to severe weather. In this paper, the element of novelty is given by reviewing these observations in a unique framework, addressing the community of NWP verification, aiming at stimulating the usage of these observations for the objective verification of high-impact weather and of specific weather phenomena. In

this context, this work proposes to establish or reinforce the bridge between the nowcasting and the impact communities on one side, and the NWP community on the other side. In particular, a closer cooperation with nowcasting groups is suggested, because of their experience in developing products for the detection of high-impact weather phenomena. This cooperation would allow for the identification of nowcasting objects and algorithms which can be used as pseudo-observations for forecast verification.

The possibility offered by non-standard types of observations generated through human activities, such as reports of severe weather, impact data (emergency calls, emergency services, insurance data), and crowdsourcing data, could be extended to many more data sources. For example, no studies referenced here made use of weather related data which can be collected automatically when a car is driving (e.g. condition of the road with respect to icing), but their potential for forecast verification

has already been foreseen (Riede et al., 2019), though not yet in a mature stage. Other possibilities are offered by new impact
data (e.g. the effect of the weather on agriculture) or by the exploitation of the huge amount of information available in the
social networks.

Performing verification of phenomena using new types of observations requires that the matching between forecasted and
observed quantities be addressed first. The paper highlights, in reviewing the referenced works, how the matching between
forecast and "new" observations has been performed by those authors, in order to provide the reader with useful indications of
when the same observations, or even other observations with similar characteristics, could be used in verification.

The assessment of the data quality and of the uncertainty associated to the observations is part of this verification. The usage
of multiple data sources is suggested as a way to take into account this issue. As we have seen, few studies follow this approach.
Finally, high-impact weather verification requires an approach to the verification problem where the exact matching between
forecast and observation is rarely possible, therefore verification naturally tends to follow fuzzy and/or spatial approaches. In
some studies, spatial verification methods already established for the verification using standard observations for rainfall (SAL,
MODE, FSS) have been applied also with non-standard observations. Following this approach, the full range of spatial methods
may be applied in this context, depending on the characteristics of the specific phenomenon and the observation used. Since
this is a relatively new field of application for the objective verification, it is not known which is the most suitable method for
each phenomenon/observations case, but the reader is invited to consider the hints provided by the authors who first entered
this realm and expand on them.

This paper does not provide an exhaustive review of all possible new observations potentially usable for the verification of
high-impact weather forecast, but it seeks to provide the NWP verification community with an organic "starter-package" of
information about new observations, their characteristics and hints for their usage in verification. This information should
serve as a basis to consolidate the practice of the verification of high-impact weather phenomena, stimulating the search, for
each specific verification purpose, of the most appropriate observations.

**Author contribution**

Conceptualization: Chiara Marsigli; Investigation: Chiara Marsigli, Raghavendra Ashrit, Caio Coelho, Elizabeth Ebert, Eric
Gilleland, Thomas Haiden, Stephanie Landman, Marion Mittermaier, Barbara Casati, Jing Chen, Manfred Dorninger;
Methodology: Chiara Marsigli, Elizabeth Ebert; Writing – original draft preparation: Chiara Marsigli; Writing – review &
editing: Chiara Marsigli, Elizabeth Ebert, Eric Gilleland, Caio Coelho, Marion Mittermaier.

**Competing interests**

The authors declare that they have no conflict of interest.



**Acknowledgments**

The first author thanks Kathrin Wapler (DWD) for constructive discussions and for reviewing an early version of the manuscript. Miria Celano (Arpae) and the colleagues of the Verification Working Group of the COSMO Consortium are acknowledged as well for useful discussions.

We are grateful to Nanette Lomarda and Estelle De Coning from WWRP for the support given to the WWRP/WGNE Joint Working Group on Forecast Verification Research (JWGFVR) of WMO.

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
