# Peer review of "Observations for high-impact weather and their use in verification"

_Natural Hazards and Earth System Sciences, 2020_

## Referee Comment (RC1) · Tomeu Rigo (Referee) · 4 Dec 2020

Title: **Observations for high-impact weather and their use in verification**

Authors: **Chiara Marsigli, Elizabeth Ebert, Raghavendra Ashrit, Barbara Casati, Jing Chen, Caio A. S. Coelho, Manfred Dorninger, Eric Gilleland, Thomas Haiden, Stephanie Landman, Marion Mittermaier**

Journal: **Natural Hazards and Earth System Sciences**

General comments

The manuscript presents a very interesting issue, which is a current main topic of interest in the verification processes around the World. The text is well-written and the language is clear. I have only some comments about general aspects:

- It is difficult for the reader understanding the link of the two selected phenomena: thunderstorms and fog. The objectives, the products and many other points are very different. If this was the main objective (to show the differences), I think that you should clarify and make a shorter text presenting the different products used for analyzing the results.
- Some of the products are very well presented but, on the contrary, other ones do not. I encourage you to make an exercise of making "uniform" them.
- One of the main differences between the products is the number of references. Some of the cases present some references and other ones only one. Having in mind that most of the presented issues have been largely studied and are easily found in the bibliography, I think that you should include more references in the poor cases
- About the lightning data, what about the lightning jump?

---

## Referee Comment (RC2) · Barbara Brown (Referee) · 8 Dec 2020

**Comments on "Observations for high-impact weather and their uses in verification" by Marsigli et al.**

**General comments**

This paper is an important contribution to the literature on forecast verification. It sets the stage for the use of new kinds of "observations" for verification and the application of verification approaches to forecasts of new, often user-relevant, phenomena (e.g., thunderstorm occurrence and impacts). Future work in this area will build on the information provided in this paper.

The paper provides citations and summaries of recent work in this area; however, it tends to be fairly focused on research in the European region, with most of the references also from the European literature. It would be useful to include additional references from other parts of the world where different experiences and knowledge exist, but the European focus is not surprising since many of the co-authors are members of the European scientific community. However, it is worth noting, for example, that U.S. researchers have made significant use of radar mosaics to evaluate convective and precipitation forecasts for the last two decades; just a couple of examples include Gilleland et al. (2009) and Roberts et al. (2013).

Another area of research that might be considered in the paper is work done by Hitchens et al. (2013) to define a "practically perfect" warning region based, for example, on point-based storm reports (e.g., for wind, hail). The method uses statistical methods to convert point observations across space into a field (a "practically perfect" forecast) that can be compared directly to a warning or the output from a model.

A relatively minor – but perhaps relevant – consideration concerns the nuance between the words "evaluation" and "verification". While we do *verification* to derive numbers that represent estimates of performance or skill (for some purpose such as monitoring changes over time), it may be more appropriate to use the word "*evaluation*" to represent many of the kinds of analyses that can be undertaken using the datasets considered in this paper. I wonder if including that terminology in the paper would help express the breadth of effort that is required to understand forecast performance, particularly for phenomena that have direct human consequences.

**Minor comments**

1. Line 36: I believe you mean "weather-related hazards" rather than "hazards weather-related".
2. Line 39: The end of this line ("weather forecast, one the main") needs editing.
3. Line 61: The last part of this line ("…combining to…") needs editing.
4. Line 76: Can you suggest other phenomena that would benefit from application of these kinds of approaches?
5. Lines 142-145: I want to note that some of the spatial methods (e.g., distance metrics, MODE) do make it relatively easy to identify and/or evaluate false alarms.
6. Line 241: Note that the human impact may still be large even if the population is sparse (e.g., in the US Midwest, localized hailstorms can destroy farm crops and have huge economic impacts).
7. Line 269: It should be made clear that the studies mentioned are a subset of those that have been undertaken world-wide to address this topic and that many other research efforts could be cited.
8. Line 289: Should "P" be "POD"? Also, what about FAR in this example?
9. Line 341: Should this be FAR rather than false alarm rate?

10. Finally, a few typographical and minor grammatical errors, scattered throughout the paper, should be corrected.

**References**

Gilleland, E., D. Ahijevych, B.G. Brown, B. Casati, and E.E. Ebert, 2009: Intercomparison of Spatial Forecast Verification Methods. *Weather Forecast*., **24** (5), 1416 - 1430, doi: 10.1175/2009WAF2222269.1.

Hitchens, N.M., H.E. Brooks, and M.P. Kay, 2013: Objective limits on forecasting skill of rare events. *Wea. Forecasting*, **28**, 525-534.

Roberts, R.D., A.S. Anderson, E. Nelson, B.G. Brown, J.W. Wilson, M. Pocernich, and T. Saxen, 2012: Impacts of forecaster involvement on convective storm initiation. *Weather and Forecasting*, **27**, 1061-1089.

---

## Author Comment (AC1) · 10 Feb 2021

Dear Referee, thank you very much for reviewing this manuscript and for providing us relevant comments and suggestions, which help us in making more focussed and more clear the paper. Hereby I reply to the points you raised.

- It is difficult for the reader understanding the link of the two selected phenomena: thunderstorms and fog. The objectives, the products and many other points are very different. If this was the main objective (to show the differences), I think that you should clarify and make a shorter text presenting the different products used for analyzing the results.

We agree with this comment, which gave us the occasion to clarify this issue in the text.

A text has been added in the introduction, motivating this choice and highlighting the differences between the two phenomena and the different usage which can be made of the different products.

- Some of the products are very well presented but, on the contrary, other ones do not. I encourage you to make an exercise of making "uniform" them.

We tried to uniform the presentation, even if this is not easy because some references come from published literature, where many details are provided, and some from "gray" literature, like presentations, which we wanted anyway to include since they can provide some ideas and hints of usage. On top, some papers present a verification, others deal with different topics but they are used as example of observations which we think can be used also in verification. This inhomogeneity of the literature reviewed is unfortunately visible in the text, as you remark, but is also the strength of the paper, we believe, in its effort to put together in a new context data and methods applied until now only to a different range of problems.

- One of the main differences between the products is the number of references. Some of the cases present some references and other ones only one. Having in mind that most of the presented issues have been largely studied and are easily found in the bibliography, I think that you should include more references in the poor cases.

We have included more references for the poor cases, where possible.

- About the lightning data, what about the lightning jump?

In none of the referenced works the lightning jump was used to discriminate between moderate and severe convection. Often the occurrence of a single lightning was taken as an indicator of convection going on, without differentiating based on the number of lightning. Some papers mention that, for verification purpose, "just one lightning" seems to be enough to detect the presence of a convective cell.

2020-362, 2020.

---

## Author Comment (AC2) · 10 Feb 2021

Dear Referee, thank you very much for reviewing this manuscript and for providing us relevant comments and suggestions, which help us in making more focussed and more clear the paper. Hereby I reply to the points you raised.

General comments

This paper is an important contribution to the literature on forecast verification. It sets the stage for the use of new kinds of "observations" for verification and the application of verification approaches to forecasts of new, often user-relevant, phenomena (e.g., thunderstorm occurrence and impacts). Future work in this area will build on the information provided in this paper.

[Figure]

We are very pleased of your opinion about this work!

The paper provides citations and summaries of recent work in this area; however, it tends to be fairly focused on research in the European region, with most of the references also from the European literature. It would be useful to include additional references from other parts of the world where different experiences and knowledge exist, but the European focus is not surprising since many of the co-authors are members of the European scientific community.

We agree with your comment, despite form the continuous international cooperation our work tends to have anyway mainly a "regional" scope, where the region is here a continent. We tried to add in review other works from non-European authors.

However, it is worth noting, for example, that U.S. researchers have made significant use of radar mosaics to evaluate convective and precipitation forecasts for the last two decades; just a couple of examples include Gilleland et al. (2009) and Roberts et al. (2013).

Thank you, these references have been added, together with a text motivating their importance in the context of the present paper.

Another area of research that might be considered in the paper is work done by Hitchens et al. (2013) to define a "practically perfect" warning region based, for example, on point-based storm reports (e.g., for wind, hail). The method uses statistical methods to convert point observations across space into a field (a "practically perfect" forecast) that can be compared directly to a warning or the output from a model.

We are very grateful for this suggestion: the paper is very stimulating and your suggestion about the usage of the "practically perfect" warning region can be highly beneficial when employing report data in verification.

A relatively minor – but perhaps relevant – consideration concerns the nuance between the words "evaluation" and "verification". While we do verification to derive numbers

that represent estimates of performance or skill (for some purpose such as monitoring changes over time), it may be more appropriate to use the word "evaluation" to represent many of the kinds of analyses that can be undertaken using the datasets considered in this paper. I wonder if including that terminology in the paper would help express the breadth of effort that is required to understand forecast performance, particularly for phenomena that have direct human consequences.

The point you raise is certainly relevant. The strength of these observations resides in the possibility they provide to verify phenomena, i.e. something manifesting itself, in this case on humans and environments. Therefore, the kinds on analyses that can be undertaken hopefully will help understanding forecast performance in a broader sense, by evaluating new aspects of the forecasts. Nowadays, however, these observations are used often only qualitatively, since it is not obvious how they should or could be compared with a model output. The main focus of the paper is actually in the effort to underline how they can become a "standard" quantitative basis against which to verify an appropriate output of the model forecast. From this point of view, the word verification (even objective verification, in an earlier version) was stressed throughout the text in order to push towards a quantitative usage, and the steps needed to derive numbers from observations which are often only text or images. Some of these considerations have been added in the text, in order to express the nuance of meanings.

Minor comments

1. Line 36: I believe you mean "weather-related hazards" rather than "hazards weather-related". Corrected.

2. Line 39: The end of this line ("weather forecast, one the main") needs editing. Done.

3. Line 61: The last part of this line ("...combining to...") needs editing. Done.

4. Line 76: Can you suggest other phenomena that would benefit from application of these kinds of approaches? A sentence suggesting other phenomena has been added

in the text.

5. Lines 142-145: I want to note that some of the spatial methods (e.g., distance metrics, MODE) do make it relatively easy to identify and/or evaluate false alarms. A sentence has been added in the text.

6. Line 241: Note that the human impact may still be large even if the population is sparse (e.g., in the US Midwest, localized hailstorms can destroy farm crops and have huge economic impacts). The sentence has been slightly modified.

7. Line 269: It should be made clear that the studies mentioned are a subset of those that have been undertaken world-wide to address this topic and that many other research efforts could be cited. A sentence has been added.

8. Line 289: Should "P" be "POD"? Also, what about FAR in this example? P has been changed in POD, it has been added a sentence about the FAR.

9. Line 341: Should this be FAR rather than false alarm rate? It is not clear from the presentation (unfortunately it is not a paper) but we agree that from the context of that test it makes sense its being a False Alarm Ratio, since the beta tester reply only when an alert is issued.

10. Finally, a few typographical and minor grammatical errors, scattered throughout the paper, should be corrected. The paper has been deeply re-read.